# ReG-NAS: Graph Neural Network Architecture Search using Regression Proxy Task

## Abstract

Neural Architecture Search (NAS) has become a focus that has been extensively researched in recent years. Innovative achievements are yielded from the area like convolutional neural networks (CNN), recurrent neural networks (RNN) and so on. However, research on NAS for graph neural networks (GNN) is still in a preliminary stage. Because of the special structure of graph data, some conclusions drew from CNN cannot be directly applied to GNN. At the same time, for NAS, the models' ranking stability is of great importance for it reflects the reliability of the NAS performance. Unfortunately, little research attention has been paid to it, making it a pitfall in the development of NAS research. In this paper, we proposed a novel NAS pipeline, ReG-NAS, which balances stability, reliability and time cost to search the best GNN architecture. Besides, for the first time, we systematically analyzed factors that will affect models' ranking stability in a given search space, which can be used as a guideline for subsequent studies. Our codes are available at `https://anonymous.4open.science/r/ReG-NAS-4D21`

## 1 Introduction

Graph neural networks (GNN) have received a lot of attention for their broad applications in social networks (Guo & Wang, 2020; Gao et al., 2021; Zhong et al., 2020), molecule properties prediction (Shui & Karypis, 2020; Ma et al., 2020; Yang et al., 2021), traffic prediction (Diehl et al., 2019; Bui et al., 2021; Zhang et al., 2021) and so on. With the goal of "faster and more accurate", people always have a pursuit to find a better structure of GNN. However, similar to neural networks like CNN and RNN, searching an ideal GNN architecture manually also is challenging. Neural architecture search (NAS) for GNN is absolutely a key point to the future development of GNN.

To design a NAS architecture, an intuitive yet primitive idea is to enumerate all models in a given search space, and evaluate each model's performance according to the metric specified by the downstream task (Ying et al., 2019; Dong & Yang, 2019; You et al., 2020). However, it is extremely time-consuming and needs a huge amount of computational resources. To make NAS more efficient, several searching methods are proposed. Most GNN NAS architectures can be divided into five classes. (1) Reinforcement-learning-based methods (Zhou et al., 2019; Gao et al., 2020; Zhao et al., 2020a), where these architectures have controllers defined as a neural network that dynamically change the parameters according to the evaluation of the performance of the generated model; (2) Bayesian-optimization-based methods (Yoon et al., 2020; Tu et al., 2019), which builds a probability distribution over sampled candidates and uses a surrogate function to test; (3) Evolution-learning-based methods (Shi et al., 2022; Li & King, 2020), among which the genetic algorithm is the most commonly used for GNN NAS frameworks (Oloulade et al., 2021). (4) Differentiable-search-based methods (Zhao et al., 2020b; Huan et al., 2021; Ding et al., 2021; Li et al., 2021b; Cai et al., 2021), which learns one or two blocks that are repeated in the whole neural network, and for GNN block is generally represented as a direct acyclic graph consisting of an ordered sequence of nodes. (5) Random-search-based methods (Gao et al., 2020; Zhao et al., 2020a; Tu et al., 2019), which generates random submodels from the search space. However, these methods are still time consuming and can take hours to days.

To reduce the search time, a popular way in NAS is to use a proxy-task, usually much smaller than the groundtruth task (e.g., Cifar10 is a proxy for ImageNet). The representativeness of the proxy-task is crucial, i.e., how similar results can be obtained from proxy-task and from groundtruth task.

One prevailing method to quantify the similarity between the two is using a "ranking correlation", which refers to the similarity between two rankings of all the networks' performance in the search space, usually uses Spearman's $\rho$ and Kendall's $\tau$ as indicators (Abdelfattah et al., 2021; Liu et al., 2020; Zela et al., 2019). The larger $\rho$ and $\tau$ are, the more similar the two rankings are, meaning the more representative of the proxy task is. By achieving large $\rho$ and $\tau$ between groundtruth ranking and prediction ranking, one can significantly improve NAS quality (Zhou et al., 2020; Chu et al., 2021; Li & Talwalkar, 2020). A lot of zero-cost or few-shot NAS works have been proposed to design a good proxy, so that the ranking correlation between proxy and groundtruth can be high within only a few training steps (Mellor et al., 2021; Dey et al., 2021; Li et al., 2021a).

In our work, we propose a proxy-task based GNN architecture search, aiming to reduce the GNN architecture search time. Similar to previous works in DNN architecture search Zhou et al. (2020); Chu et al. (2021); Li & Talwalkar (2020); Mellor et al. (2021); Dey et al. (2021); Li et al. (2021a), we also use ranking correlation to evaluate the performance of our proposed proxy-based NAS. In addition, ranking correlation can be used to quantify the *ranking stability* of the networks by computing $\rho$ and $\tau$ between different repetitions of the same training pipeline. Large $\rho$ and $\tau$ imply that the variation of the model's relative ranking is small, i.e., the ranking is stable. We clarify two metrics that will be used hereafter:

- **Ranking Correlation**: Correlation of two network rankings on two different tasks, e.g., ground truth task and proxy task, or proxy task one and proxy task two; quantified by $\rho$ and $\tau$.

- **Ranking Stability**: Correlation of two rankings on the same task but of two repetitions, either the same or different initialization and training hyperparameters. Also quantified by $\rho$ and $\tau$.

Observing the two metrics, we found an interesting phenomenon: in a GNN search space, *the ranking stability for classification tasks can be much lower than for regression tasks*. For classification groundtruth tasks, the ranking correlation between two repetitions of all GNN architectures in the same search space can be as low as 0.57, while for regression tasks, the ranking correlation between two repetitions can be up to 0.99. Inspired by this observation, together with a recent regression-based proxy task in CNN, GenNAS (Li et al., 2021c), we propose a self-supervised **regression-based proxy task** for GNN NAS. We observe that using our proposed regression-based proxy task, both the ranking stability and ranking correlation are higher. In addition, regression-based proxy task converges faster than classification groundtruth task, thus reducing the search time. However, generate a *representative* proxy task is non-trivial. There is a rich study for proxy task generation or selection for DNN NAS but no related research on GNN NAS. The only regression-based DNN NAS work, GenNAS, can automatically search for a good proxy task, but requires knowing 20 architectures with groundtruth ranking (Li et al., 2021c). This is a strong premise and still can be time consuming when targeting a new search space or dataset, i.e., at least 20 architectures must be well-trained and then ranked.

To address the above challenges, we proposed a novel NAS method, using **Re**ression-based proxy task for **G**raph **N**eural **A**rchitecture **S**earch, **ReG-NAS**. We summarize our contributions as follows:

- ReG-NAS is the *first* GNN NAS using *regression-based proxy task*. We propose a GNN NAS pipeline that can transform a groundtruth classification task to a regression proxy task, which leads to much higher ranking stability and faster convergence.

- We systematically study the ranking stability and ranking correlation under various training environments, and uncover the fact that *directly searching on classification groundtruth task is unreliable* because of the low ranking stability. This observation challenges one common practice in NAS that, as long as the ranking correlation between proxy and groundtruth task is high, it is regarded as an effective proxy.

- We propose a simple yet effective proxy task to guide GNN NAS, which does not require groundtruth labels but only one well-trained GNN model as proxy task generator. The generator is not necessarily the best GNN but can be *any* GNN within the search space. Using the proposed proxy task, we turn the groundtruth classification problem into regression, leading to much higher ranking stability and faster search.

## 2 RELATED WORK

### 2.1 GRAPH NEURAL NETWORKS

Graph is a kind of data structure that defines a set of nodes and their relationships (Waikhom & Patgiri, 2021). A graph is a set of $V$ nodes and a set of $E$ edges, with optional labels $\mathbf{y}$ or features $\mathbf{x}$ attached to nodes, links, or the whole graph. Therefore, a graph can be represented as $G = (V, E)$. Graph Neural Networks (GNN) is a type of Deep Neural Networks (DNN) that is suitable for analyzing graph-structured data. If we use $\mathbf{x}_v$ and $\mathbf{x}_{uv}$ to represent node $v$'s and link $(u, v)$'s feature vectors, use $\mathbf{h}_v$ and $\mathbf{h}_{uv}$ to represent node $v$'s and link $(u, v)$'s hidden representations in GNN, the GNN's message passing and update process can be described as:

$$\mathbf{h}'_v = f_{\text{node}}\left(\mathbf{h}_v, \sum_{u \in \mathcal{N}(v)} \mathbf{h}_{uv}, \mathbf{x}_v\right) \tag{1}$$

$$\mathbf{h}'_{uv} = f_{\text{edge}}\left(\mathbf{h}_u, \mathbf{h}_v, \mathbf{x}_{uv}\right) \tag{2}$$

Where $\mathcal{N}(v)$ denotes the number of in-neighbor nodes of node $v$, and $f_{\text{node}}$ and $f_{\text{edge}}$ are message passing functions that gather information from node's neighborhood and previous layers.

GNN can be classified according to variants of graphs, downstream tasks, learning methods and so on. Main types of graphs includes undirected/directed graph, heterogeneous graph, dynamic graph, attributed graph and so on. Downstream tasks usually include classification task and regression task, each of which can be subdivided into graph-level, link-level and node-level problems. In our work, we mainly focus on undirected graphs with graph-level tasks.

### 2.2 GRAPH NEURAL ARCHITECTURE SEARCH AND GRAPHGYM

Graph Neural Architecture Search (GNN NAS) means automatically find the best GNN models for targeted tasks (Oloulade et al., 2021). Like NAS for other neural networks, GNN NAS samples an architecture from a predefined search space, then the NAS network will evaluate the performance of sampled architecture as a feedback returned to the search algorithm.

GNN NAS can be categorized according to search space, search algorithm (see Section 1) and performance evaluation (Oloulade et al., 2021). Recent studies for GNN NAS (Zhou et al., 2019; Gao et al., 2020; Tu et al., 2019; Shi et al., 2022; Huan et al., 2021; Li et al., 2021b; You et al., 2020) have not only achieved promising performance for many applications of GNNs but also showed potential as a unanimous approach to constructing GNN models.

Among all GNN NAS frameworks, we want to introduce GraphGym (You et al., 2020) in detail. In Graph-Gym, a GNN design space consists of 12 design dimensions for intra-layer design, inter-layer design and learn-

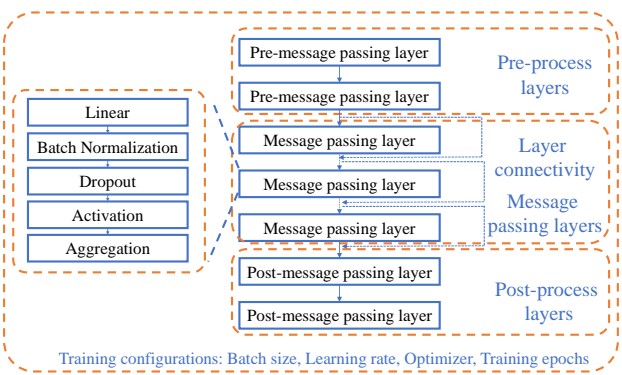

Figure 1: Structure of and Design Space of GNN

ing configuration. A single GNN layer has a sequence of modules:(1) Linear layer $\mathbf{W}^{(k)}\mathbf{h}_u^{(k)} + \mathbf{b}^{(k)}$; (2) batch normalization $\text{BN}(\cdot)$ (Ioffe & Szegedy, 2015);(3) dropout operation $\text{DROPOUT}(\cdot)$ (Srivastava et al., 2014); (4) nonlinear activation function $\text{ACT}(\cdot)$; (5) aggregation function $\text{AGG}(\cdot)$. Therefore, the $k$-th GNN layer can be defined as:

$$\mathbf{h}_v^{(k+1)} = \text{AGG}\left(\left\{\text{ACT}\left(\text{DROPOUT}\left(\text{BN}\left(\mathbf{W}^{(k)}\mathbf{h}_u^{(k)} + \mathbf{b}^{(k)}\right)\right)\right), u \in \mathcal{N}(v)\right\}\right) \tag{3}$$

where $\mathbf{h}_v^{(k)}$ is the $k$-th layer embeddings of node $v$, $\mathbf{W}^{(k)}, \mathbf{b}^{(k)}$ are trainable weights. For inter-layer design, Graphgym gives three ways to connect GNN layers: STACK (directly stack multiple GNN layers) (Welling & Kipf, 2016; Velickovic et al., 2017); SKIP-SUM (residual connections) (He et al., 2016) and SKIP-CAT (concatenate embeddings in all previous layers) (Huang et al., 2017). GraphGym also adds Multilayer Perceptron (MLP) layers before/after GNN message passing. Training Configurations includes batch size, learning rate, optimizer type and training epochs. The overview of GraphGym's design space and structure is shown in Fig 1. In our works, we utilize GraphGym as a basic framework and propose our new NAS architecture by modifying it.

## 2.3 Generic Neural Architecture Search via Regression (GenNAS)

Generic Neural Architecture Search (GenNAS) is the a CNN and RNN based NAS framework that uses self-supervised regression proxy task instead of classification for NAS (Li et al., 2021c). Compared to other NAS frameworks, it has several advantages: (1) By using regression as the self-supervised proxy task, it is **downstream-agnostic** to the specific downstream tasks. (2) It has **near-zero training cost**, which is highly efficient for neural architecture search.

Here we mainly focus on CNN regression architectures of GenNAS, as shown in Fig 2(a). GenNAS constructs a Fully Convolutional Network (FCN) (Long et al., 2015) by removing the final classifier of a CNN, and then extract the FCN's intermediate feature maps from multiple stages. The number of stages is denoted as $N$. If the input tensor is $I$, each stage's feature map tensor is $\mathcal{F}_i$, the synthetic signal is $\mathcal{F}_i^*$, the regression pipeline will reshape $\mathcal{F}_i$ into $\hat{\mathcal{F}}_i = M_i(\mathcal{F}_i)$, and compute MSE loss defined as $\mathcal{L} = \sum_{i=1}^{N} \mathbf{E}[(\mathcal{F}_i^* - \hat{\mathcal{F}}_i)^2]$ during training process. GenNAS will rank models' performance according to final MSE values, and select the model with the lowest MSE value as the best. GenNAS uses Ranking Correlation (See Section 1) for NAS evaluation.

## 3 Proposed ReG-NAS

### 3.1 The barriers to generate proxy task

As mentioned in Section 1, ReG-NAS uses a regression based proxy task to search for GNN structures. However, different from grid-data like images, which can use the combination of signals with different frequency but the same *shape* (i.e., data dimension) as proxy task, generate proxy task for GNN is much harder.

First, even in the same dataset, different graphs usually have different topology structures. Therefore, for graph datasets, we should generate proxy task for each graph individually, and there is no so-called "global" signal (Li et al., 2021c) in the process of generating proxy task.

Second, although according to spectral graph theory (Shuman et al., 2013), any graph signal can be projected on the eigenvectors of the Laplacian Matrix $L$ and the "frequency" of each eigenvector is the corresponding eigenvalue, we cannot simply use these vectors as our proxy task. For graphs, Laplacian Matrix only contains a graph's structural information, while many graphs also have other non-structural information such as node features and edge features. Direct linear combination of Laplacian Matrix's eigenvectors will lose graphs' original information and weaken NAS performance. Therefore, in our generating process, we should aggregate both structural and non-structural information into proxy task.

### 3.2 Proposed ReG-NAS Pipeline

In ReG-NAS, we propose a simple, yet effective proxy task generator for graph datasets, as shown in Fig 2(b). In a given search space with number of post-process layers equals to 2, we first randomly select a GNN architecture, then use this architecture to train the dataset for $k$ epochs. In the end we use this well-trained model for inference, and extract model's hidden node feature from the first Post-process layer as our proxy task $\mathcal{F}_p$. The reason why we set the number of Post-process layers into 2 is to make sure that all graphs' proxy tasks are in the same shape. For example, if the input graph $G$'s original node feature is $\mathcal{I} \in \mathbb{R}^{n \times d_0}$ ($n$ is the number of nodes), and in the message passing layer $i$, node feature is $\mathcal{F}_i \in \mathbb{R}^{n \times d_1}$, the first post-process layer will reshape $\mathcal{F}_i$ into $\mathcal{F}_p \in \mathbb{R}^{1 \times d_p}$, and $\mathcal{F}_p$ is the proxy task for graph. The first post-process layer uniformly converts node feature's

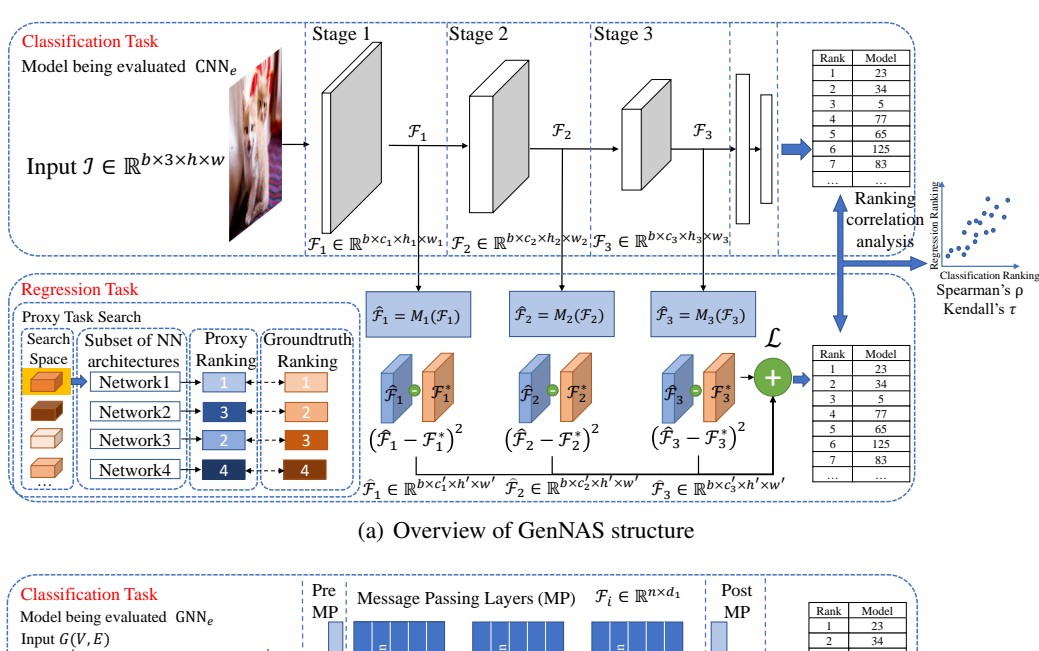

(a) Overview of GenNAS structure

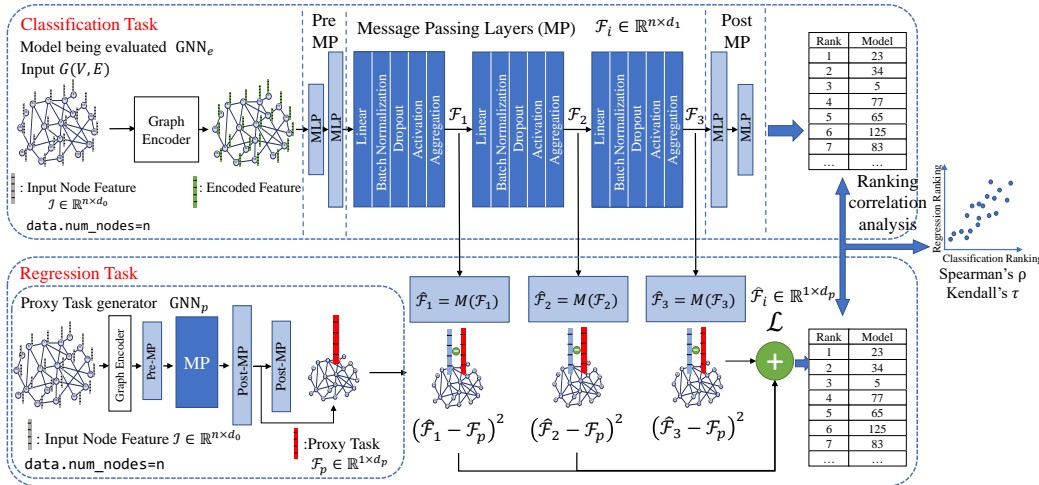

(b) Overview of ReG-NAS structure

Figure 2: Overview of GenNAS structure (a) and ReG-NAS structure (b). For ReG-NAS, this figure only shows the cases when groundtruth task is classification. In fact, ReG-NAS is also applicable for regression-based groundtruth task, which is almost the same as classification-based groundtruth task architecture searching pipeline.

shape into $1 \times d_p$ despite the variance of $d_0$ and $d_1$, which facilitates the proxy training process in the next.

The way we generate proxy task **does not need to know model's performance ranking, even a subset of the search space.** As the final goal of NAS is to rank GNN's *relative* performance, the generator's performance wouldn't affect the final result as long as the generated proxy task is informative (See Section 4.1.2). In fact, from the experiments discussed later, we will find that the selection of model has little effect on the final results. This is different from the way that GenNAS did (Li et al., 2021c): In GenNAS, before generating proxy task, we need to know the relative performance ranking of a subset (e.g 20) of the neural architectures in NAS search space, which is a strong assumption and thus is not practical when being applied to new datasets or tasks.

After the proxy task is generated, we will attach it to the graph dataset. In the regression training process, ReG-NAS will extract node feature $\mathcal{F}_i$ from each Message Passing layer, and reshape it into $\hat{\mathcal{F}}_i \in \mathbb{R}^{1 \times d_p}$ in order to match the shape of $\mathcal{F}_p$. We use pooling method (SUM, MEAN, MAX)

to reshape $\mathcal{F}_i$. The evaluation metric is the final MSE loss between $\hat{\mathcal{F}}_i$ and $\mathcal{F}_p$. In the end we will compute Ranking Coefficient (Spearman's $\rho$ and Kendall's $\tau$) between **proxy (regression-based) ranking** and **groundtruth (classification-based or regression-based) ranking**.

In summary, the whole process of ReG-NAS contains several steps below:

1. Randomly select a GNN model from the search space as proxy task generator $\text{GNN}_p$;
2. Generate proxy task $\mathcal{F}_p \in \mathbb{R}^{1 \times d_p}$;
3. Select a model to be evaluated $\text{GNN}_e$ from the search space;
4. Extract Message Passing layer's node feature $\mathcal{F}_i \in \mathbb{R}^{n \times d_1}$, reshape it into $\hat{\mathcal{F}}_i \in \mathbb{R}^{1 \times d_p}$, compute MSE loss $\mathcal{L} = \sum_{i=1}^{N} \mathbf{E}[(\hat{\mathcal{F}}_i - \mathcal{F}_p)^2]$;
5. Evaluate and rank model's performance according to final $\mathcal{L}$ value.

## 4 EXPERIMENTS

Table 1: Basic information and hyper-parameter configurations for **ogbg-molhiv** and **ZINC**

| Dataset | ogbg-molhiv | | ZINC | |
|---|---|---|---|---|
| # Graphs | 41,127 | | 249,456 (12,000 for subset) | |
| # Nodes per graph | 25.5 | | 23.15 | |
| # Edges per graph | 27.5 | | 24.9 | |
| Task type | Binary-classification | | Regression | |
| Metric | ROC-AUC | | MAE | |
| Pipeline | Groundtruth (classification) | Proxy (Regression) | Groundtruth (Regression) | Proxy (Regression) |
| Base learning rate | 0.0007 | 0.075 | 0.0006 | 0.0009 |
| Batch size | 128 | 128 | 128 | 128 |
| Dropout | 0 (False) | 0 (False) | 0 (False) | 0 (False) |
| Loss function | Cross entropy | MSE | MSE | MSE |
| Optimizer | ADAM | SGD | ADAM | ADAM |
| Node encoder | Atom | Atom | Atom | Atom |
| Edge encoder | Bond | Bond | None | None |
| # Post-mp layers | 2 | 2 | 2 | 2 |
| Train Epochs | 100 | 80 | 100 | 80 |

To fully evaluate the performance of ReG-NAS as well as analyze the factors that will affect GNN ranking stability, we conduct 3 types of experiments: Ranking stability analysis, Effectiveness evaluation and Efficiency evaluation. In these experiments we use `ogbg-molhiv` (Hu et al., 2020) as classification-based dataset, and use `ZINC` (Gómez-Bombarelli et al., 2018) as regression-based dataset. The basic information of these datasets and hyper-parameter configurations are listed in Table 1. Our search space contains 216 GNN models, as shown in Table 2.

All hyper-parameters (base learning rate, optimizer, batch size etc.) are optimized to ensure that the model converges at an optimal rate. In our experiment, the learning rate is annealed via cosine decay to 0 in order to reduce the variance between multiple independent training runs (Loshchilov & Hutter, 2016). And for `ZINC`, we use its subset which contains 12,000 graphs with

Table 2: GNN Search Space for experiments

| **Variable** | **Range** |
|---|---|
| # Message Passing layers | 2, 3 |
| Stage | STACK, SKIP-SUM, SKIP-CAT |
| Inner Layer Dimension | 32, 64, 128, 256 |
| Activation | ReLU, SWISH, PReLU |
| Aggregation | MEAN, MAX, SUM |

10,000 train graphs, 1,000 test graphs and 1,000 validation graphs in our experiment to reduce training cost. The reason why we set proxy training epochs equals to 80 is that we find that all models' loss converges at about 80 epochs, thus there's no need to train extra 20 epochs.

### 4.1 GNN RANKING STABILITY ANALYSIS

In this section we aim to find factors that will affect GNN ranking stability and try to evaluate the stability of our proposed NAS pipeline. Therefore, we first test GNN ranking stability on two groundtruth task (classification task on `ogbg-molhiv`, regression task on `ZINC`), then we test ranking stability on our NAS pipeline. At the same time, we will also discuss how proxy task will

affect the proxy ranking stability. For a single experiment, we repeat the training and evaluation of all architectures 3 times, then compute Ranking Stability among them. For different repetitions of the same task, their initialization is different.

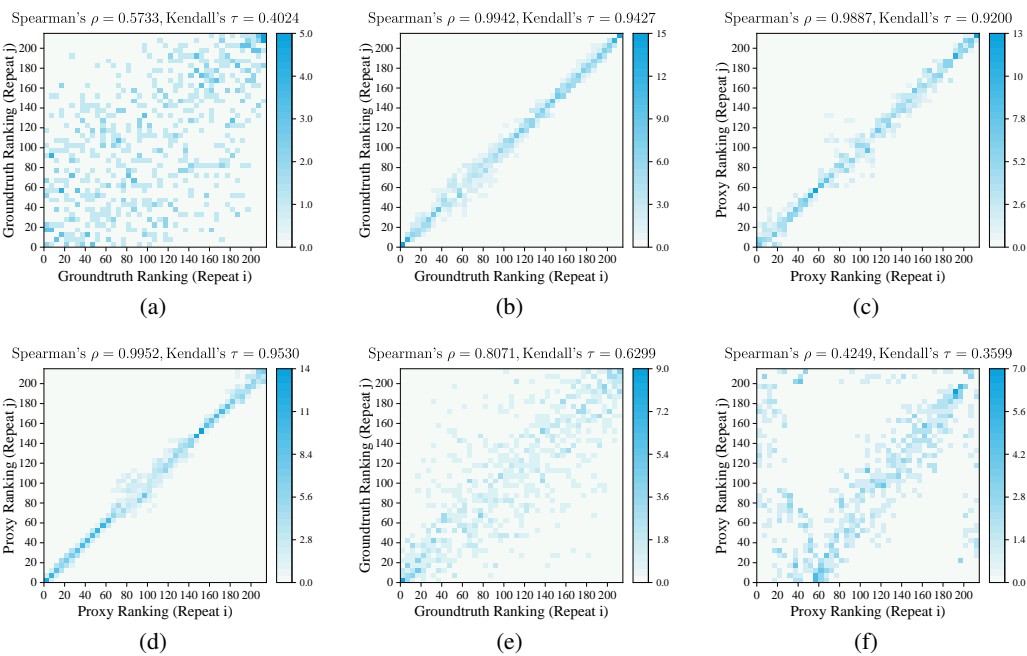

Figure 3: Groundtruth task's and Proxy task's Ranking Stability analysis. (a) Classification-based groundtruth task's Ranking Stability on ogbg-molhiv (Use ROC-AUC as ranking metric); (b) Regression-based task's Ranking Stability on ZINC (Use MAE as ranking metric); (c) Proxy task's Ranking Stability on ogbg-molhiv (Use MSE as ranking metric); (d) Proxy task's Ranking Stability on ZINC (Use MSE as ranking metric); (e) Classification-based groundtruth task's Ranking Stability on ogbg-molhiv (Use Loss as ranking metric); (f) Proxy task's Ranking Stability on ogbg-molhiv (Use random-generated vectors as proxy task)

The Ranking Stability analysis between two repetitions of the same experiment are shown in Fig 3. The $x$-axis represents the ranking of $i$th experiment, and the $y$-axis represents the ranking of $j$th experiment. For example, if a model ranks 3rd in the $i$th experiment and ranks 5th in the $j$th experiment, then the model's coordinate is (3, 5). We make Ranking Stability analysis for every 2 experiments and place all points on the same figure, represented as a heat map. The heat map will show the density of the points and therefore reveals the pattern of task's Ranking Stability for a training pipeline with a given configuration. We also compute $\rho$ and $\tau$ from the heat map.

### 4.1.1 GROUNDTRUTH RANKING AND PROXY RANKING STABILITY ANALYSIS

In the situation when we don't modify ranking metrics on groundtruth ranking and proxy ranking, and use proposed method to generate proxy task, as shown in Fig 3(a)-3(d), from the result we can clearly see that regression-based ranking are more stable than classification-based ranking, despite using groundtruth training pipeline or proxy training pipeline. This phenomenon may due to the choice of downstream task as well as ranking metric. For classification task, in the training process, the variable being directly optimized is loss value, while the final ranking metric is ROC-AUC (for `ogbg-molhiv`); At the same time, for regression task, in the training process the variable being directly optimized is still loss value, but the final ranking metric is MAE (for `ZINC`). An intuitive explanation is that, network performance for regression-based task is evaluated directly on the regression loss; the network performance for classification based task, on the other hand, is evaluated on ROC-AUC, which is an indirect metric. To further validate our hypothesis, based on `ogbg-molhiv`, we rank models according to their final loss value, and analyze Ranking Stability,

as shown in Fig 3(e). From the result we can find that compared to "ROC-AUC-based" ranking, "loss-based" ranking's $\rho$ and $\tau$ are much better, which affirms our hypothesis.

### 4.1.2 THE RELATIONSHIP BETWEEN THE CHOICE OF PROXY TASK AND PROXY TASK'S RANKING STABILITY

Although all the experiments mentioned above illustrate regression-based rankings are more stable, this may not be applicable to all cases. For example, if we use randomly generated vectors as our proxy task and use it to rank models, we will find it's $\rho$ is only 0.424, even it is a regression-based ranking, as Fig 3(f) shows. From the results we can find many points fall in the upper left and lower right corners of the figure, which means that the relative rankings of these models differ a lot in the two repeated experiments. This is an anomaly even for classification cases.

To explain this phenomenon, we should know the way how the Deep Learning optimization works. Deep Learning optimization is trying to globally optimize a function by using local gradient information (Bottou & Bousquet, 2007), which means if a learning problem is characterized by non-informative gradients, then no deep learning architecture will be able to learn it. Back to the topic, clearly random-generated vectors cannot provide informative gradients, due to which it is not surprising that a model's ranking changes drastically in different repetitions. Therefore, before we draw a conclusion that regression-based rankings are more stable, it should based on a premise that **the learning problem should be informative or reasonable**.

### 4.2 REG-NAS PERFORMANCE EVALUATION

### 4.2.1 EFFECTIVENESS OF REG-NAS

We compute Ranking Correlation between groundtruth rankings and proxy rankings to show the effectiveness of ReG-NAS. To shed light on how the variance of groundtruth tasks and proxy tasks affect the effectiveness of ReG-NAS, we conduct 8 types of experiments. For `ogbg-molhiv`, we conduct 5 types of experiments, the difference among them is the way how proxy task is generated: (1) Use Random-selected Model as proxy task generator (RM); (2) Use model with the best groundtruth performance in the search space ("Golden" Model) as proxy task generator (GM); (3) Use model with the worst groundtruth performance in the search space ("Poorest" Model) as proxy task generator (PM); (4) Use Random-generated Vectors as proxy task (RV); (5) Use Laplacian Matrix's eigenvectors as proxy task (LE). For LE-based pipeline, as the shape of Laplacian Matrix's eigenvectors $(n \times 1)$ are not equal to $1 \times d_p$, so the reshape process is different from other proxy training pipeline, as Fig 4

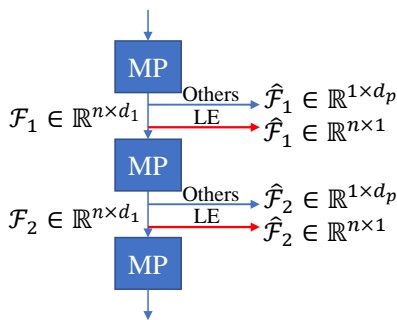

Figure 4: Comparison between LE pooling and other pooling process

shows. For `ZINC`, we conduct 3 types of experiments: (1) RM-based pipeline; (2) GM-based pipeline; (3) PM-based pipeline. For each groundtruth task and proxy task, we repeat the experiment three times, and compute the average value of Spearman's $\rho$ and Kendall's $\tau$ among them as final results, as shown in Table 3.

Table 3: Proxy-Groundtruth Ranking Correlation analysis

| Datasets | ogbg-molhiv | | | | | ZINC | | |
|---|---|---|---|---|---|---|---|---|
| Groundtruth task type | Classification | | | | | Regression | | |
| Proxy task | RM | GM | PM | RV | LE | RM | GM | PM |
| Proxy task's shape | $1 \times d_p$ | $1 \times d_p$ | $1 \times d_p$ | $1 \times d_p$ | $n \times 1$ | $1 \times d_p$ | $1 \times d_p$ | $1 \times d_p$ |
| Spearman's $\rho$ | **0.362** | **0.367** | **0.356** | -0.064 | -0.038 | **0.421** | **0.435** | **0.443** |
| Kendall's $\tau$ | 0.260 | 0.259 | 0.227 | -0.041 | -0.026 | 0.303 | 0.314 | 0.324 |

From the results we can draw three conclusions: **(1) Random-generated vectors and Laplacian Matrix's eigenvectors are not suitable for proxy task.** As mentioned before, for proxy training

pipeline, the learning problem should be informative and reasonable (Section 4.1.2). Therefore, Random-generated vectors cannot be used as proxy task. Meanwhile, from Section 3.1 we know that Laplacian Matrix's eigenvectors doesn't contain graph's non-structural information, which is the key reason to the poor performance of LE pipeline. **(2) RM/GM/PM-based pipeline reached an ideal Spearman's $\rho$ and Kendall's $\tau$,** which can be used to approximate the performance of GNN structure. **(3) The choice of models as proxy task generator has little effects on the final results, regardless the type of groundtruth task.** For `ogbg-molhiv`, the difference between the $\rho$ of GM-based pipeline and PM-based pipeline is 0.011, which is in the fluctuation range of $\rho$ (See Section 4.2.2); For `ZINC`, the PM-based pipeline's $\rho$ is even higher than GM-based pipeline's, which further proves that we don't have to select "Golden" model as task generator.

### 4.2.2 EFFICIENCY OF REG-NAS

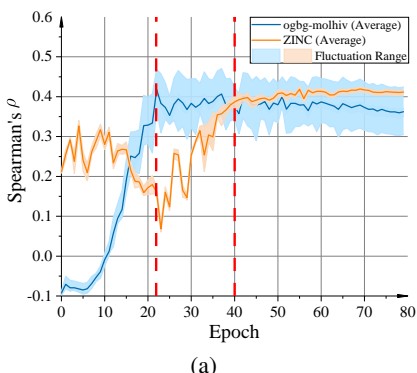

| Dataset | ogbg-molhiv | | ZINC | |
|---|---|---|---|---|
| Task Type | Ground truth | Proxy | Ground truth | Proxy |
| Required Training epochs | 100 | 20 | 100 | 40 |
| Average Single-epoch training time (s) | 20.52 | 24.43 | 2.03 | 2.22 |
| Speedup | 4.2× | | 2.3× | |

(a)                                                                 (b)

Figure 5: ReG-NAS convergence speed analysis. (a) Spearman's $\rho$ convergence curve, $\rho$ is the Ranking Correlation between proxy ranking at epoch $n$ and final groundtruth ranking at epoch 100. (b) ReG-NAS speedup compared to Groundtruth training time.

Here we compute Spearman's $\rho$ between proxy ranking at epoch $n$ and final groundtruth ranking at epoch 100 (constant), and draw the relationship between $\rho$ and epoch. Since we repeat each experiment 3 times, we use Filled Area Graph to show the fluctuation range of rho at each epoch, as shown in Fig 5(a). According to the figure we can find that compared to `ZINC`, `ogbg-molhiv`'s $\rho$ fluctuates more drastically, which possibly due to the instability of groundtruth ranking. Meanwhile, for `ogbg-molhiv`, $\rho$ converges at about 20 epochs, and for `ZINC`, $\rho$ converges at about 40 epochs. Fig 5(b) illustrates the speedup of ReG-NAS during the search of best GNN architecture by testing the average single-epoch training time of groundtruth pipeline and proxy pipeline. For `ogbg-molhiv`, ReG-NAS can save up to 76.2% training time; For `ZINC`, ReG-NAS can save 56.5% training time.

To sum up, ReG-NAS is downstream-agnostic (applicable on both classification task and regression task), stable, effective and efficient GNN NAS architecture. By using ReG-NAS, we can approximate GNN's relative performance in a short time, which is especially useful for searching GNN in a large search space.

## 5 CONCLUSION

In this work, we proposed ReG-NAS, a GNN NAS architecture which uses regression proxy task. It has several advantages: (1) Stable. The Ranking Stability (Spearman's $\rho$) between two repetitions can reach up to 0.99; (2) Downstream-agnostic. It is applicable to both classification task and regression task; (3) Effective. It has high proxy-groundtruth ranking similarity which can be work as a reference of GNN's relative performance; (4) Efficient. Compared to traditional NAS searching method, it can save up to 76.2% of training time. At the same time, for the first time, we analyzed the factors that will affect GNN ranking stability, which provides a new insight of designing a stable GNN.

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
