# OpenReview forum: "ReG-NAS: Graph Neural Network Architecture Search using Regression Proxy Task"
_ICLR.cc/2023/Conference — Submitted to ICLR 2023_

### Official Review · Reviewer_bV9M · 2022-10-24

**Confidence:** 4
**Correctness:** 2
**Technical Novelty And Significance:** 2
**Empirical Novelty And Significance:** 3
**Recommendation:** 3

**Clarity, Quality, Novelty And Reproducibility:**

This paper proposes an interesting perspective for graph NAS, but the quality and the clarity of the paper are in a pretty immature state.

The reproducibility should be good, considering an anonymous link is provided.

**Strength And Weaknesses:**

Pros:
1. The study about the stability of using classification and regression tasks to estimate GNN architecture performance is interesting and useful, which can inspire future work in graph NAS.
2. The paper is clearly written and easy to follow.

Cons:
Though I find the paper studies an interesting perspective of graph NAS, the current experiments are highly insufficient with the following concerns:
1. I’m concerned that only using SSL tasks to estimate the architecture performance may not work well for GNNs. Specifically, it is well known that the performance of architectures heavily relies on the downstream task, e.g., see [1] for an empirical study and [2] for a theoretical justification. For example, what if the SSL task is fundamentally different from the downstream tasks?
[1] Design space for graph neural networks, NeurIPS 2020.
[2] How neural networks extrapolate: from feedforward to graph neural networks, ICLR 2021.

2. The experiments in Section 4 are not convincing enough. The authors only use two molecule datasets, neglecting that graph datasets cover a wide range of domains, e.g., social networks, citation networks, financial networks, etc. Different data sources can cause the graph to behave fundamentally differently. It is doubtful that the same conclusion holds in other areas.

3. The Spearman correlation coefficient and Kendall Tau shown in Table 3 are not very high, so I wonder whether they can support a good NAS method. Moreover, the authors do not show the quantitative results of the pipeline for the final downstream task, i.e., the real effectiveness of the proposed method is not validated. Besides, I would suggest the authors additionally show the correlation of the top-performing architectures instead of all architectures in the search space, considering that the ultimate goal of NAS is to find good architectures rather than evaluation all possible architectures.

4. The statement in Section 4.1.2 regarding using random labels seems to be in contradiction with NAS results in other domains, e.g., in computer vision [3]. The authors may need to explore deeper into why random labels do not work for graph tasks.
[3] Neural architecture search with random labels, CVPR 2021.


**Summary Of The Paper:**

This paper proposes a new graph neural architecture search pipeline named Reg-NAS. Specifically, the authors find that regression tasks are more reliable than classification tasks in estimating architecture performance and advocate that a regression self-supervised proxy task be used to estimate the architecture performance.

**Summary Of The Review:**

See above

---

> ### Author Response · Authors · 2022-11-17
> **About random labels in NAS**
>
> Thank you so much for your time and valuable comments.
>
> We noticed the paper *Neural architecture search with random labels* proposed a way of searching neural network by using random labels. However, the final metric to judge whether a model is good is to evaluate their speed of convergence, instead of validation accuracy. In fact, from table 1 in *Neural architecture search with random labels*, we can find that if we use random labels to search architecture and ranking models according to their validation accuracy, the best model searched in this way has the lowest performace compared to other methods, which does not contradict with our conclusion.

---

### Official Review · Reviewer_i2b5 · 2022-10-24

**Confidence:** 5
**Clarity, Quality, Novelty And Reproducibility:** The paper is well written and the sou…
**Correctness:** 2
**Technical Novelty And Significance:** 3
**Empirical Novelty And Significance:** 3
**Recommendation:** 5

**Strength And Weaknesses:**

### Strengths:
1. REG-NAS only needs one well-trained proxy model, unlike the previous work GenNAS which needs at least 20 models for CNN and RNN.

2. The proposed proxy task requires fewer epochs to converge than the ground-truth task which reduces the search time.

### Weaknesses:
1. My most concern is the integrity of the paper. From Table 3, we can observe that the Spearman or Kendall Correlation Coefficient is not good enough (e.g., 0.362/0.260 for ogbg-molhiv and 0.421/ 0.303 for ZINC). I have no idea that the metric of the proxy regression task can really replace the downstream metric (AUC, MAE). Thus, in my opinion, the authors should show how the proposed REG-NAS improves the search efficiency of the NAS methods (e.g., RL, EA) on the downstream metric.

2. There is the same question for another contribution: ranking stability. This paper takes a lot of space to talk about the proxy metric is stable and the ground-truth classification metric (AUC) is unstable under different initializations. However, in the practical application, we focus more on ground-truth metrics. If the ground-truth metric is unstable, why do we need a stable proxy metric to replace the ground-truth metric?

In my opinion, our problem is a GNN NAS problem and I think the proxy task may focus on improving search efficiency for GNN with ground-truth metric. However, it seems that this paper doesn't give sufficient proof for this aspect.


**Summary Of The Paper:**

This paper first proposes a proxy regression task, which is called REG-NAS, to replace the metric of the ground-truth task and evaluate the GNN architecture performance. The proxy regression task and its metric leading lead to higher ranking stability and faster search.

**Summary Of The Review:**

This paper provides a proxy task approach to simplified evaluation metrics for GNN NAS. However, it doesn't provide sufficient proof that the proxy task improves search efficiency for GNN NAS.

---

### Official Review · Reviewer_TB9W · 2022-10-25

**Confidence:** 4
**Correctness:** 2
**Technical Novelty And Significance:** 2
**Empirical Novelty And Significance:** 2
**Recommendation:** 3

**Clarity, Quality, Novelty And Reproducibility:**

Please see Strength And Weaknesses.
It is good to see code is included.

**Strength And Weaknesses:**

I have several concerns about this paper:

1. Finding a proxy task for GNN architecture search is not appealing, because GNN is typically not deep (especially in your search space) and its training is much cheaper compared with training a deep CNN or a transformer. Only moderate search time is needed to find a good GNN reliably (e.g., with evolutionary search). It is questionable whether reducing search time via a proxy task is worth the extra problems it brings.

2. The description of the proxy pipeline is not clear enough. For example, for each candidate in the search space, is it trained to minimize the MSE error, or is it trained with the original loss function (MSE error is only used as the final evaluation)?

3. The proxy design needs further justification. Feature maps from different message-passing layers are compared to the same F_p of the proxy generator GNN, which is counterintuitive.  Analysis from GNN's perspective should be added.

4. The experimental results are not convincing enough. The correlation in Table 3 is poor and cannot support the proposed proxy task. Small search space, small graph size, only 2 datasets.

**Summary Of The Paper:**

This paper designs a proxy task for GNN architecture search. Specifically, a GNN is randomly selected from the search space as the proxy generator. For each candidate architecture in the search space, MSE error between its feature map and that of the proxy generator is computed and used to rank this candidate. This paper then analyses the stability and correlation of such ranking method using two graph-level datasets.

**Summary Of The Review:**

Please see Strength And Weaknesses

---

> ### Author Response · Authors · 2022-11-17
> **Rebuttal about concerns**
>
> Thank you so much for your time and valuable comments. We will address your concerns accordingly.
> > Finding a proxy task for GNN architecture search is not appealing, because GNN is typically not deep (especially in your search space) and its training is much cheaper compared with training a deep CNN or a transformer.
>
> The cost of training GNN are not only determined by the depth of GNN. Other factors like inner layer dimension, type of dataset s(especially graph based datasets, which often contain thousands or millions of graphs) will greatly affect GNN's training time. Our GNN NAS mainly focus on graph-level datasets, which actually consumes a lot of training time, and we think it is neccessary to enhance the time of searching GNN.
>
> >The description of the proxy pipeline is not clear enough. For example, for each candidate in the search space, is it trained to minimize the MSE error, or is it trained with the original loss function (MSE error is only used as the final evaluation)?
>
> For regression based *groundtruth* task, loss function is MSE. Therefore, there's no difference between loss value and MSE error value. For regression based *proxy*  task, loss function is the MSE error between proxy task and hidden feature from message passing layer. In our experiment, we minimize loss value during the training and rank models according to their final MSE value.
>
> > The proxy design needs further justification. Feature maps from different message-passing layers are compared to the same $F_p$ of the proxy generator GNN, which is counterintuitive. Analysis from GNN's perspective should be added.
>
> We humbly accept the advice and will find a better solution of generating proxy task in our future work.

---

### Official Review · Reviewer_xCbv · 2022-10-27

**Confidence:** 5
**Correctness:** 2
**Technical Novelty And Significance:** 2
**Empirical Novelty And Significance:** 2
**Recommendation:** 3

**Clarity, Quality, Novelty And Reproducibility:**

Some claims may be inaccurate. The novelty is not significant compared previous method (see Concerns for the details).

**Strength And Weaknesses:**

Strength:

The idea of using regression as a proxy task is interesting.

Concerns:

- Why the ranking stability for classification tasks can be much lower than for regression tasks? Any justification?

- The authors claim to be the first to analyze factors that will affect models’ ranking stability for GNNs. However, to the best of my knowledge, this has been systematically studied in SGAS [1] which is not discussed. I believe a dedicated section for the comparison and discussion with SGAS is needed.

- The authors propose to generate the proxy task by a randomly selected GNN architecture. However, it would cause significant biases toward similar architectures as the selected GNN architecture. For instance, if a GNN with sum aggregations is selected. The search algorithm will favor GNNs with sum aggregations because of the expressiveness of GNNs.

- At the 3rd step of ReG-NAS, a model is selected to be evaluated GNNe from the search space. I wonder how the model is selected. Is it brute force, RL-based, evolution-based, or differentiable-based?

- The final results of obtained architectures on ogbg-molhiv and ZINC are not reported. There is no comparison with SOTA models.

[1] Li, G., Qian, G., Delgadillo, I.C., Muller, M., Thabet, A. and Ghanem, B., 2020. Sgas: Sequential greedy architecture search. In Proceedings of the IEEE/CVF Conference on Computer Vision and Pattern Recognition (pp. 1620-1630).

[2] Xu, K., Hu, W., Leskovec, J. and Jegelka, S., 2018. How powerful are graph neural networks?. arXiv preprint arXiv:1810.00826.


======== post rebuttal ========

Thanks for the authors' reply. However, my main concern is that the method is only evaluated on a small search space with a brute-force search approach. It is nonclear if it can be generalized to a large search space. I keep my rating unchanged.

**Summary Of The Paper:**

This work proposed to search GNN architectures by searching on a proxy regression task. The proxy regression task is generated by regressing some intermediate input features to output features from a pretrained GNN model which is randomly selected from the search space. The results are evaluated based on Ranking Coefficient Spearman’s ρ and Kendall’s τ.

**Summary Of The Review:**

The manuscript can be improved for future submission.

---

> ### Author Response · Authors · 2022-11-17
> **Rebuttal about concerns**
>
> Thank you so much for your time and valuable comments. We will address your concerns accordingly.
> > Why the ranking stability for classification tasks can be much lower than for regression tasks? Any justification?
>
> We gave an explanation in section 4.1.1.
>
> > About Comparison with SGAS
>
> In SGAS, what it is discussed is the selection stability that measures the movement of the operation distribution, which is different from what we discussed in this paper. In this paper we mainly focus on how the change of random seeds (which will determine the initialization of the parameter settings, models' combination in a batch etc.) will affect the final models ranking in a given search space, which is not discussed in SGAS.
>
> > At the 3rd step of ReG-NAS, a model is selected to be evaluated GNNe from the search space. I wonder how the model is selected. Is it brute force, RL-based, evolution-based, or differentiable-based?
>
> In our paper, with the goal of obtain more results in a relatively short time, our search space is not large (contains 216 models). Therefore, we directly evaluate all models (maybe is the "brutal force" you indicated) and rank them according to their final performace.
>
> > The final results of obtained architectures on ogbg-molhiv and ZINC are not reported. There is no comparison with SOTA models.
>
> Because of the difference of the search space, we cannot give a comparison with SOTA models. In other words, the design space of GNN models may not include the architecture of SOTA models, thus the best model may not achieve the highest accuracy compared with SOTA models. Plus, our goal is try to make our proxy ranking similar to groundturth ranking, which means that we consider our model is good as long as it can achive a high $\rho$ with groundturth ranking. Therefore, maybe the comparison with SOTA model in not neccessary.

---

### Decision · Program_Chairs · 2023-01-20

**Decision:**

Reject

**Justification For Why Not Higher Score:**

Several concerns remain unsolved after the rebuttals.

**Justification For Why Not Lower Score:**

N/A

**Metareview: Summary, Strengths And Weaknesses:**

The paper proposed a proxy task for GNN architecture search. The proposed approach balances stability, reliability, and time cost to search for GNN architecture.


Strength:
+ Overall, the research topic is interesting and the proposed approach has great potential.

Weakness:
- One of the key concerns is that the method is not tested in a realistic setting. It's only evaluated on a small search space with a brute-force search approach. Therefore, it's unclear if it can be useful for practical GNN problems.
- Some of the design choices are not well-justified. More analysis and discussions are needed.
- The experiment results are not convincing and sufficient to support the main claims of the paper.

Please refer to reviewers' comments for more details.





**Summary Of Ac-Reviewer Meeting:**

N/A